# Optimization of Extraction Process, Structure Characterization, and Antioxidant Activity of Polysaccharides from Different Parts of *Camellia oleifera* Abel

**DOI:** 10.3390/foods11203185

**Published:** 2022-10-12

**Authors:** Shiling Feng, Min Tang, Zhengfeng Jiang, Yunjie Ruan, Li Liu, Qingbo Kong, Zhuoya Xiang, Tao Chen, Lijun Zhou, Hongyu Yang, Ming Yuan, Chunbang Ding

**Affiliations:** 1College of Life Science, Sichuan Agricultural University, Ya’an 625014, China; 2Institute of Agro-Products Processing Science and Technology, Sichuan Academy of Agricultural Sciences, Chengdu 610066, China

**Keywords:** *Camellia oleifera*, Box−Behnken design, physicochemical property, antioxidant activity

## Abstract

The flowers, leaves, seed cakes and fruit shells of *Camellia oleifera* are rich in bioactive polysaccharides, which can be used as additives in food and other industries. In this study, a Box−Behnken design was used to optimize the extraction conditions of polysaccharides from *C. oleifera* flowers (P-CF), leaves (P-CL), seed cakes (P-CC), and fruit shells (P-CS). Under the optimized extraction conditions, the polysaccharide yields of the four polysaccharides were 9.32% ± 0.11 (P-CF), 7.57% ± 0.11 (P-CL), 8.69% ± 0.16 (P-CC), and 7.25% ± 0.07 (P-CS), respectively. Polysaccharides were mainly composed of mannose, rhamnose, galacturonic acid, glucose, galactose, and xylose, of which the molecular weights ranged from 3.31 kDa to 128.06 kDa. P-CC had a triple helix structure. The antioxidant activities of the four polysaccharides were determined by Fe^2+^ chelating and free radical scavenging abilities. The results showed that all polysaccharides had antioxidant effects. Among them, P-CF had the strongest antioxidant activity, of which the highest scavenging ability of DPPH^•^, ABTS^•+^, and hydroxyl radical could reach 84.19% ± 2.65, 94.8% ± 0.22, and 79.97% ± 3.04, respectively, and the best chelating ability of Fe^2+^ could reach 44.67% ± 1.04. Overall, polysaccharides extracted from different parts of *C. oleifera* showed a certain antioxidant effect, and could be developed as a new type of pure natural antioxidant for food.

## 1. Introduction

*Camellia oleifera* Abel. is an evergreen shrub or small tree of the *Theaceae* family, which is widely distributed in the middle and lower reaches of the Yangtze River of China, also found in Japan, India, and Southeast Asia [1]. *C. oleifera* seed oil is known as “Oriental olive oil” due to its high content of unsaturated fatty acids and multitude of natural antioxidants, including squalene, phytosterol and polyphenols [2,3,4,5]. The cultivation acreage of *C. oleifera* in China had reached approximately 4.5 million hectares by 2021 (National Forestry and Grass Administration, 2022). In order to achieve a high yield of *C. oleifera*, it is necessary to prune the leaves and thin flowers of trees annually. Pruning 1/3-1/2 or even more than 1/2 of the branches and leaves promotes the main economic traits, such as fruit yield of *C. oleifera*. In addition, people also take off the excess flowers to ensure the tree has enough nutrients for *C. oleifera* fruits. The pruning and thinning operation of *C. oleifera* will produce a large number of agronomic by-products annually. These agronomic by-products are usually piled up or discarded directly in the field and even burned to pollute the environment. Some studies have shown that the leaves and flowers have various bioactive substances such as saponins, polysaccharides, and polyphenols, which have antioxidant, hypoglycemic, and anti-inflammatory activities. *C. oleifera* leaf polysaccharide showed an excellent antioxidant effect on iron chelating, hydroxyl radical and DPPH^•^ (2,2-diphenyl-1-picrylhydra-zyl) scavenging due to its low molecular weight, higher content of mannose and uronic acid, which could be developed into a new type of antioxidant [6]. The *Camellia* fruit shells and seed cakes are the main by-product of woody edible oil production. Approximately 8 million and 4 million tons of *C. oleifera* fruit shells and seed cakes, respectively, are produced annually after oil extraction [7]. These byproducts of oil extraction are usually used to produce fertilizers, detergents, activated carbon, and protein feed [8,9]. Zhang et al. found a polysaccharide from *C. oleifera* fruit shells that expressed a hypoglycemic effect. They believed this function may be related to the inhibition of α-glucosidase by acidic polysaccharide [10]. The polysaccharide from *C. oleifera* seed cakes showed hypoglycemic activity in mice, which may be the effect of mannose and rhamnose [11]. These results indicated that the biological activity of *C. oleifera* polysaccharides was closely related to the molecular weight, uronic acid, and monosaccharide composition. Therefore, the deep processing by-products of agronomic and oil extraction significantly increase the additional value of *C. oleifera*.

Plant polysaccharide is a natural active substance derived from crops, fruits, vegetables, and herbs. It has become a research hotspot in food, medical treatment, cosmetics, and health care products due to the advantages of low toxicity and multiple biological activities. Many studies showed that natural polysaccharides from plants have the effects of antioxidant, anti-tumor, antibacterial, hypoglycemic, and immunomodulatory [12,13,14]. The water-soluble polysaccharide obtained from bamboo shoot by-products could treat antibiotic-associated diarrhea (AAD) in mice and are considered to be natural ingredients that can improve gut health [15]. Furthermore, Lanzhou lily polysaccharide could promote macrophage proliferation, secret regulatory factors (nitric oxide, interleukin-6, and tumor necrosis factor-α). Therefore, it can be applied to the treatment of immunoregulatory drugs [16]. Recently, in order to improve high-value *C. oleifera*, various polysaccharides from leaves, flowers, fruit shells and seed cakes have been separated. However, there are few systematic studies on polysaccharides from different parts of *C. oleifera*, and the polysaccharides from *C. oleifera* flowers have not been reported. Therefore, based on the Box−Behnken design, this study optimized the hot-water extraction process (extraction temperature, time, and liquid–solid ratio) of polysaccharides from *C. oleifera* flowers, leaves, seed cakes, and fruit shells. Moreover, we also explored the structure of each polysaccharide by detecting monosaccharide composition, molecular weight, UV–visible spectroscopy, Fourier transform infrared (FTIR), and Congo red. Four different methods (DPPH^•^, ABTS^•+^ (2,2′-Azinobis [3-ethylbenzothiazoline-6-sulfonic acid]-diammonium salt), Hydroxyl radical and Fe^2+^ chelate) were used to test and compare the antioxidant effects of each polysaccharide. These results will provide a reference for by-products of agronomic and oil extraction to develop pure natural food antioxidants.

## 2. Materials and Methods

### 2.1. Raw Materials

The flowers, leaves, and fruit shells of *C. oleifera* were collected in autumn from a *C. oleifera* breeding base in Ya’an City, Sichuan Province (30° N, 103° E). The seed cakes were provided by Tai Shi Biological Company, Ya’an City, Sichuan Province. All experimental materials in this study were identified by Professor Chunbang Ding. All materials were dried in a vacuum oven and crushed into powder sieved by 40 mesh.

### 2.2. Chemicals and Reagents

Deionized water was used to prepare the solution required for the test. Petroleum ether (60–90 °C), acetone, alcohol, trichloromethane, n-butyl alcohol, sulfuric acid, phenol, glucose, ascorbic acid and other reagents were purchased from Chengdu Kelong Chemical Reagent Factory (Chengdu, China). DPPH^•^ and ABTS^•+^ were purchased from Sigma-Aldrich Chemical Company (St. Louis., MO, USA). All the chemicals used were reagent-grade. Methyl alcohol and acetonitrile were chromatographic-grade.

### 2.3. Experimental Design

Hot water extraction is the most traditional and straightforward extraction method, which can protect glycosidic bonds from damage to the maximum extent. A Box−Behnken design (BBD) avoids the defects of other experimental designs and ensures the effectiveness of research under extreme conditions [17]. Therefore, in this study, extraction temperature, extraction time, and liquid–solid ratio were used as single factors with a BBD to establish the best hot water extraction conditions for polysaccharides. The yield of polysaccharide (YP) was the response value, and the calculation method was as follows:YP (%) = m/M(1)
where m was the weight of crude extract; M was the weight of dry powder.

### 2.4. Optimization of Polysaccharides Extraction Process by BBD

The dried raw materials were refluxed with petroleum ether, acetone and ethanol to remove pigments, lipids, and oligosaccharides. Dry powder was used to extract polysaccharides based on BBD. Briefly, Under different extraction temperatures, the solid powder was mixed with different volumes of water and extracted at different times. The extract was centrifuged at 4000 rpm/min for 10 min and concentrated to 1/10 of the original volume by a rotary evaporator. Then, add 5 times the volume of anhydrous ethanol to the concentrate overnight at 4 °C. The precipitate produced by the action of ethanol is collected and redissolved again. After that, the protein was removed by Sevag method, and small molecular substances were removed by dialysis bag with retention of 3500 Da. Finally, crude polysaccharides were obtained by freeze-drying.

### 2.5. Structure Characterization

#### 2.5.1. Determination of Chemical Composition

The total carbohydrate content was determined by the phenol-sulfuric acid method with glucose as standard [18]. In short, 100 μL polysaccharide solution or glucose solution was mixed with 100 μL phenol solution and 500 μL sulfuric acid, then, reacted at room temperature for 10 min. The absorbance was measured at 490 nm. The protein content was determined by the Coomassie brilliant blue G-250 with bovine serum albumin as standard [19]. Briefly, 200 μL polysaccharide solution or glucose solution was mixed with 1 mL bovine serum albumin, then, reacted at room temperature for 30 min. The absorbance was measured at 595 nm.

#### 2.5.2. Determination of Molecular Weight

Determination of molecular weight of polysaccharides was by high-performance gelpermeation chromatography (HPGPC) [20]. The detector was a refractive index detector (RID) at 35 °C; chromatographic column: PLaquagel-OH (7.5 × 300 μm) column (Agilent Technologies, Santa Clara, CA, USA) at 30 °C; the sample injection volume was 20 μL; 0.1 mol/L (M) NaCl as mobile phase at a flow rate of 0.5 mL/min. The concentration of P-CF, P-CL, P-CC, and P-CS for detection was 5 mg/mL. Before detection, the sample was filtered through a membrane (0.22 μm, Millipore, Burlington, MA, USA).

#### 2.5.3. Monosaccharide Composition Analysis

The monosaccharide composition was determined by high-performance liquid chromatography (HPLC) [20]. In short, 4 mg/mL P-CF, P-CL, P-CC, and P-CS solution was mixed with 1 mL trifluoroacetic acid and placed in an ampoule bottle, and hydrolyzed at 100 °C for 6 h. Trifluoroacetic acid was removed by rotary evaporation and redissolved in 3 mL of deionized water. Then 500 μL acid hydrolysis solution or monosaccharide standard solution was added to the test tube, followed by addition of 625 μL PMP-ethanol solution (0.5 M), 1 mL NaOH solution (0.3 M), 70 °C water bath treatment for 40 min, and 5 mL chloroform extraction three times. The upper water phase was taken and filtered with 0.22 μm filter before determination. An Agilent 1260 HPLC system with a Zorbax SB-C18 column (150 × 4.6 mm, 5 μm) was used to detect monosaccharide composition by a UV-vis DAD detector at 250 nm. The mobile phase consisted of 0.05 M PBS (pH = 6.9) and acetonitrile at a flow rate of 1 mL/min.

#### 2.5.4. UV–Visible Spectroscopy

UV-1800 (Daojin Instrument Limited Company, Suzhou, China) was used for full-band scanning of the four polysaccharide solutions (0.25 mg/mL) at 200–400 nm.

#### 2.5.5. Fourier Transform Infrared (FTIR) Spectroscopy

The dried polysaccharide and KBr were fully ground into powder at a ratio of 1:50. Then, these powders were made into uniform transparent sheets and then analyzed by infrared spectrometry via the Shimadzu 8400 spectrophotometer (Shimadzu, Kyoto, Japan) at 4000–400 cm^−1^.

#### 2.5.6. Congo Red Analysis

The Congo red test was used to detect whether the polysaccharide had a triple helix structure, slightly modified according to the reported method [21]. Then, 1 mL polysaccharides (1 mg/mL) solution was mixed with Congo Red solution (1.5 ml, 100 μM). Then, 2.5 mL NaOH (1 M) was added to distribute the NaOH concentration in 0.05–0.5 M. The mixture was reacted at room temperature for 20 min. Finally, the maximum absorption wavelength was determined by UV-1800 (Suzhou Daojin Co., Ltd., China) in the range of 400–600 nm.

### 2.6. Determination of Antioxidant Activity

Four methods determined the antioxidant functions of P-CF, P-CL, P-CC, and P-CS. The DPPH^•^ scavenging ability was slightly changed according to Xiang et al. [22]. The absorbance was measured at 517 nm after the dark reaction of 180 μL DPPH^•^ (0.4 mM) and 20 μL test solution for 30 min; ABTS^•+^ scavenging capacity was determined by Li et al. [23]. The absorbance was measured at 734 nm after the dark reaction of 180 μL ABTS^•+^ reaction system (7.4 mM ABTS and 2.6 mM K_2_S_2_O_8_) and 20 μL test solution for 6 min; the hydroxyl radical scavenging ability was determined according to Wang et al. [24]. Then, 100 μL test solution was reacted with 500 μL FeSO4, 200 μL salicylic acid–ethanol solution, 500 μL H_2_O_2_, and 200 μL deionized water at 37 °C for 1 h. Then, the absorbance was measured at 510 nm; the determination of Fe^2+^ chelating ability referred to Yu et al. [25]. The absorbance was measured at 560 nm after the dark reaction of 50 μL test solution and 150 μL FeSO_4_-phenanthroline solution for 1 h, EDTA-2Na as the positive control.

### 2.7. Statistics and Analysis

Design-Expert 11 was used to Optimize the response from the model. The applicability of the model was determined by the lack of fit, determination coefficient (R^2^), and significance of the model. The significance of the model analysis was determined on the basis of *p* < 0.05. GraphPad Prism 8.2.1 (San Diego, California, USA) was used to analyze the experimental data. All the data were expressed as: mean ± standard deviation (SD), *n* = 3.

## 3. Results and Discussions

### 3.1. Optimization of Extraction Conditions

#### 3.1.1. Effects of Single Factors on the Yield of Polysaccharide

Radar plots reflect the effect of independent factors on YP of P-CF, P-CL, P-CC and P-CS (Figure 1). Figure 1(1a–4a) shows the effects of extraction temperature on the yields of P-CF, P-CL, P-CC and P-CS. With the increase in extraction temperature, the yields of four polysaccharides showed a gradual upward trend. When the temperature increased from 60 to 75 °C, the yields of P-CF and P-CL increased by 3.74% ± 0.38 (Figure 1(1a)) and 1.96% ± 0.1 (Figure 1(2a)), respectively. When the temperature continued to rise to 85 °C, the yield of P-CS increased by 2.58% ± 0.11 (Figure 1(4a)); when the temperature was ranging from 65 to 80 °C, the yield of P-CC increased by 3.45% ± 0.33 (Figure 1(3a)). After that, the yields of P-CF, P-CL, P-CC, and P-CS began to decline with the temperature increasing continuously. This may have been due to the increased diffusion coefficient of polysaccharides under higher temperature [26]. High temperature also causes polysaccharide degradation by breaking glycosidic bonds to destroy its structure and affect biological activities (antioxidant, anti-inflammatory, hypoglycemic, and so on). Therefore, 75 °C was selected as the center point of P-CF and P-CL, while 80 and 85 °C were, respectively, the center points of P-CC and P-CS for response surface design.

The effects of extraction time on the yields of P-CF, P-CL, P-CC and P-CS were shown in Figure 1(1b–4b). A longer extraction time promoted the proportion of polysaccharides dissolved in the solvent. When the extraction time of P-CF and P-CL were 100 min, the yields reached the maximum value of 8.46% ± 0.38 (Figure 1(1b)) and 8.71% ± 0.19 (Figure 1(2b)), respectively. P-CC and P-CS obtained the best extraction effects at 110 min and 120 min, respectively, and the yields were 8.48% ± 0.14 (Figure 1(3b)) and 6.47% ± 0.13 (Figure 1(4b)). The continuous extension of extraction time led to the decrease in YP, which can be explained as that under high-temperature extraction conditions, the longer extraction time may lead to the lower thermal stability of polysaccharides and cause the degradation of polysaccharides [27]. Therefore, 100, 100, 110, and 120 min were selected as the center points of P-CF, P-CL, P-CC, and P-CS for BBD test.

Another important factor affecting polysaccharide yield was liquid–solid ratio. As shown in Figure 1(1c–4c), the increase in liquid–solid ratio was beneficial to improve the yields of P-CF, P-CL, P-CC and P-CS. When the liquid–solid ratio increased from 15 to 25 mL/g, the yield of P-CF (Figure 1(1c)) reached the highest point (8.13% ± 0.08). When the liquid–solid ratio increased from 5 to 20 mL/g, the yield of P-CL (Figure 1(2c)) reached the highest point (7.48% ± 0.25). When the solid–liquid ratio gradually increased to 30 mL/g, the yields of P-CC (Figure 1(3c)) and P-CS (Figure 1(4c)) were the highest, 9.88% ± 0.30 and 6.19% ± 0.03, respectively. This is because the increase in the proportion of solvent can promote its entry into the cell, so more polysaccharides dissolve into the solvent [28]. However, when the liquid–solid ratio was further increased, the yields of the four polysaccharides began to decrease. This may be because the liquid–solid ratio reached a certain level, and the polysaccharide has been extracted completely. Since then, continuously increasing the liquid–solid ratio will cause more impurities (monosaccharides, oligosaccharides, etc.) to dissolve into the extract which may reduce the concentration of polysaccharide in the unit extract, and it will also bring difficulties to the subsequent concentration work. Therefore, 25 and 20 mL/g were selected as the centers of P-CF and P-CL for the BBD test, while 30 mL/g was selected as the center point of P-CC and P-CS. According to the above results, the single factor level of each material for the BBD test was determined (Table 1).

#### 3.1.2. Effect of Independent Variable Parameters on Response Variables

Table 2 shows YP under the random combination of different independent variables. The yields of the four polysaccharides were distributed in 5.24–9.35% (P-CF), 5.84–7.63% (P-CL), 6.4–8.27% (P-CC), and 4.65–6.94% (P-CS), respectively. Overall, P-CF had the highest polysaccharide yield compared with the other three polysaccharides.

#### 3.1.3. ANOVA Analysis of BBD and Model Fitting

Based on the BBD experiments, a second-order polynomial regression model for the yields of P-CF (Y_1_), P-CL (Y_2_), P-CC (Y_3_) and P-CS (Y_4_) was established (Table 3). The quadratic regression equation of the full variable was as follows:Y_1_ = 9.08 + 0.6336X_1_ + 0.7008X_2_+ 0.5090X_3_ − 1.14X_1_X_2_ – 0.2276X_1_X_3_ − 0.0187X_2_X_3_ – 0.5904X_1_^2^ – 0.7647X_2_^2^ − 0.3415X_3_^2^
(2)
Y_2_ = 7.06 + 0.1622X_1_ + 0.1647X_2_+ 0.2633X_3_ − 0.1244X_1_X_2_ − 0.2935X_1_X_3_ + 0.1580X_2_X_3_ − 0.4072X_1_^2^ + 0.0016X_2_^2^ − 0.0942X_3_^2^
(3)
Y_3_= 8.14 + 0.3083X_1_ + 0.1391X_2_+ 0.2719X_3_ − 0.3067X_1_X_2_ − 0.3899X_1_X_3_ + 0.0547X_2_X_3_ − 0.5289X_1_^2^ − 0.0578X_2_^2^ − 0.1699X_3_^2^
(4)
Y_4_ = 5.67 + 0.5252X_1_ + 0.3734X_2_+ 0.2198X_3_ + 0.2788X_1_X_2_ − 0.0583X_1_X_3_ + 0.3340X_2_X_3_ + 0.0850X_1_^2^ − 0.0394X_2_^2^ − 0.3049X_3_^2^
(5)
where X_1_, X_2_ and X_3_ represent extraction temperature, extraction time, and liquid–solid ratio, respectively.

The models of the four polysaccharides were significant, while any lack of fit was not significant, indicating that the models were reliable (Table 3) [29]. R^2^ and adjusted R^2^ values can reflect the predictability of the model [30]. R^2^ of Y_1_, Y_2_, Y_3_, and Y_4_ values were 0.99, 0.89, 0.98 and 0.97; adjusted R^2^ values were 0.97, 0.76, 0.96, and 0.93, respectively. This showed that for Y_1_, Y_3_ and Y_4_, more than 90% of the response variables could be explained by the model, indicating that the fitting degree of the model was excellent [31].The response variables could be explained by more than 70% via the model in Y_2_, indicating that the fitting degree was good. These models can be used to navigate the design space.

#### 3.1.4. Analysis of the Response Surface Design

Response surface and contour plots can reflect the influence of the interaction of independent factors on response variables. The slope of the response surface plot and the shape of the contour plot can reflect whether the interaction between variables is obvious or not. A steep response surface plot and elliptic contour plot mean the interactions between the variables are significant, while a smooth response surface plot and circular contour plot mean otherwise [32]. According to Table 3, the interaction between X_1_X_2_ was significant for P-CF; for P-CL, the effect of X_1_X_3_ interaction on response variables was remarkable; for P-CC, the interaction between X_1_X_2_ and X_1_X_3_ was significant; for P-CS, the interaction between X_1_X_2_ and X_2_X_3_ was obvious. Therefore, to highlight the effect of the interaction of independent factors on the response variables, we only analyzed the response surface and contour maps of the above independent factors with a significant interaction.

The contour plots Appendix A showed an ellipse, indicating that for fixed X3, the interaction of X1X2 had a significant influence on the yields of P-CF, P-CC and P-CS. At any temperature, the yields of P-CF (Figure 2(1a)) and P-CC (Figure 2(3a)) increased and then decreased slowly with extraction time increasing. Likewise, at any given extraction time, as the temperature increased, the yields of P-CF (Figure 2(1a)) and P-CC (Figure 2(3a)) increased significantly and eventually tended to be flat. This decrease could be explained as a long extraction time or high temperature may causing polysaccharide degradation [26]. In addition, the slope of Figure 2(1a) was steeper than Figure 2(3a), and the change of the yield of P-CF was more obvious. Combined with the results in Table 3, it could be concluded that for P-CF and P-CC, the influence of the extraction time (X_2_) of the former (P-CF) on YP was more significant than that of the latter (P-CC). For P-CS, we found that when the extraction time was prolonged from 80 to 160 min, and the extraction temperature was increased from 75 to 95 °C, the yield of P-CS (Figure 2(4a)) continued to increase. However, a longer time and high-temperature extraction may lead to changes in the structure of polysaccharides, such as the cleavage of glycosidic bonds, thereby affecting activity (antioxidant, hypoglycemic, anti-inflammatory, and so on).In the end, we selected the extraction temperature and time as 95 °C and 160 min for the best extraction conditions of P-CS.

The contour plots in Appendix A were close to ellipticity, indicating the effect of X_1_X_3_ on the yields of P-CL (Appendix A) and P-CC (Appendix A) were remarkable when X_2_ was fixed. Under any liquid–solid ratio, the yields of P-CL and P-CC were promoted significantly with increasing extraction temperature. When the extraction temperature was fixed, increasing liquid–solid ratio could promote the yields of P-CL and P-CC to some extent. Nevertheless, with a further increase in the liquid–solid ratio, the increase in yields was not obvious. This may be because, increasing the solid–liquid ratio could improve the contact area between sample and solvent which was beneficial to avoid incomplete extraction. However, an excessive liquid–solid ratio had little effect on polysaccharide yield and more solvents will mean trouble for subsequent concentration. Therefore, an appropriate increase in the extraction temperature and solid–liquid ratio could effectively increase the yields of P-CC and P-CL. Figure 2(3b) had steeper plots than the contour of Figure 2(2b), and combined with the dates in Table 3, it could be concluded that the effect of X_1_X_3_ on the yield of P-CC was more obvious than that of P-CL.

The contour plot (Appendix A) showed an elliptical shape which indicated that the interaction between extraction time and liquid–solid ratio was advantageous to enhance the yield of P-CS. It can be seen from Figure 2(4c) that under any fixed extraction time or liquid–solid ratio, the yield of P-CS increased first and then decreased with the change of the liquid–solid ratio or extraction time.

In accordance with the results of Figure 2, Appendix A and Table 3, the optimal extraction conditions were predicted as follows: the extraction temperatures of P-CF, P-CL, P-CC and P-CS were 72.43, 73.47, 78.17, and 95 °C; the extraction times were 123.05, 110, 130, and 160 min; the liquid–solid ratios were 32.86, 30, 40, and 42.2 mL/g. Under these conditions, the predicted polysaccharide yields of P-CF, P-CL, P-CC, and P-CS were 9.43%, 7.65%, 8.45%, and 7.09%, respectively. In order to facilitate the experiment, the predicted extraction conditions were adjusted, and three experiments were repeated to verify the BBD model (Table 4). After adjustment, the polysaccharide yields of P-CF, P-CL, P-CC, and P-CS were 9.32% ± 0.11, 7.57% ± 0.11, 8.69% ± 0.16, and 7.25% ± 0.07. There was no significant difference between the actual and predicted values, indicating that the BBD model was reliable and could be used for subsequent experiments.

### 3.2. Chemical Component, Molecular Weight and Monosaccharide Composition Analysis

According to the results in Table 5, P-CF and P-CL had higher total carbohydrate content (59.48% ± 0.59 and 64.37% ± 0.66, respectively). P-CC (3.38% ± 0.2) had the lowest protein content, while P-CF (17.66% ± 0.1) had the highest content, indicating that P-CF may be a protein-binding polysaccharide [33].

The molecular weight of four polysaccharides ranged from 3.31 kDa to 128.06 kDa, which showed an uneven distribution, indicating P-CF, P-CL, P-CC and P-CS were heterogeneous polysaccharides.

As shown in Table 5 and Figure 3, the P-CF, P-CL, and P-CS were all composed of mannose, rhamnose, galacturonic acid, glucose, galactose, xylose, and arabinose, while their molar ratios of monosaccharide composition were different. The polysaccharides extracted from *C. oleifera* leaves by Feng and co-workers had similar monosaccharide composition, indicating that the polysaccharides in this study may have the same biological activity as those obtained by Feng et al. [6]. P-CF and P-CL were mainly composed of galactose and xylose. P-CS was mainly composed of galacturonic acid, galactose and xylose. P-CC lacked arabinose and was mainly composed of glucose and xylose, which suggests that P-CC may have different biological activities compared with the other three polysaccharides.

### 3.3. The Results of UV–Visible Spectroscopy

Absorption at 260 and 280 nm indicated that polysaccharides contained nucleic acids and proteins [34]. As shown in Figure 4, P-CF, P-CL and P-CS had obvious absorption peaks at 280 nm. However, no protein absorption peak was found in P-CC. This could be explained by the minimum protein content of P-CC in Table 5.

### 3.4. FT-IR Spectra

The infrared spectra of P-CF, P-CL, P-CC, and P-CS are shown in Figure 5. The strong and wide peaks near 3355.91 cm^−1^ were due to the tensile vibration of O-H. The weak absorption peak at about 2927.74 cm^−1^ was attributed to C-H stretching vibration [35]. The stretching vibration of C=O in the esterified carboxyl groups caused absorbance peaks around 1733.88 cm^−1^ [36]. The absorption peak appeared at about 1616.24 cm^−1^, indicating that there was C=O [35]. The characteristic absorption peak of the C-H bond was around 1236.29 cm^−1^. Absorption peaks in the range of 400–900 cm^−1^ (P-CF: 759.90 cm^−1^ and 578.60 cm^−1^; P-CS: 767.62 cm^−1^ and 636.47 cm^−1^; P-CL and PCC: 763.76 cm^−1^ and 636.47 cm^−1^) indicated a pyranose ring [34]. The above analysis suggests that P-CF, P-CL, P-CC, and P-CS all had typical absorption peaks of polysaccharides.

### 3.5. The Result of Congo Red Test

Congo red as an acid dye can combine with polysaccharide with triple helix structure. Compared with Congo red (control group), the maximum absorption wavelength of the complex will shift red. [21]. According to Figure 6, in the range of 0–0.5 M NaOH, with the increase in alkali concentration, the maximum absorption wavelengths of P-CF, P-CL, and P-CS all showed a downward trend, only P-CC showed increased first and then stabilized within a certain range. According to Li et al. [37], with the increase in NaOH concentration, the polysaccharide–Congo red complex had obvious shift red, which proved that the polysaccharide exists in a triple helix conformation in aqueous solution. Our study showed that P-CC was a polysaccharide with a relatively stable triple helix structure.

### 3.6. Antioxidant Activity

#### 3.6.1. DPPH^•^ Scavenging Activity

DPPH^•^ is a stable nitrogen-centered free radical, which is widely used to evaluate the antioxidant activity of plant extract compounds in vitro. According to Figure 7a, when the polysaccharide concentration was 3 mg/mL, the DPPH^•^ scavenging activities of P-CF, P-CL, P-CC, and P-CS reached 73.15% ± 1.59, 42.81% ± 0.8, 11.31% ± 1.83 and 42.79% ± 2.95, respectively. However, as the polysaccharides concentration increased further, the DPPH^•^ scavenging ability of P-CS increased significantly, it was higher than that of P-CL and P-CC, but lower than that of P-CF and Vc. The IC_50_ values of P-CF and P-CL were 1.53 ± 0.26 and 3.16 mg/mL ± 0.75, indicated that P-CF had an obvious effect on scavenging DPPH^•^.

Hydroxyl and carboxyl groups in antioxidants provide hydrogen to reduce DPPH^•^ [38]. Monosaccharide composition has a certain impact on antioxidant activity. The contents of rhamnose and mannose were positively correlated with DPPH^•^ scavenging ability. Especially, rhamnose was the most significant determinant factor associated with antioxidant activity [39,40]. Liu et al. reported that the citrus pectin with higher rhamnose content showed more vigorous antioxidant activity [41]. Moreover, Fimbres-Olivarria et al. extracted three sulfated polysaccharides from *Navicula* sp., and found that the polysaccharide rich in rhamnose could scavenge more DPPH^•^ [42]. Furthermore, Zhong et al. [43] found that polysaccharide with higher galactose content also had stronger DPPH^•^ scavenging ability. Combined with the data in Table 5, the total proportions of mannose, rhamnose and galactose in monosaccharide composition of P-CF, P-CL, P-CC and P-CS were 41%, 37.66%, 21.72% and 34.92%, respectively, indicating that P-CF has stronger DPPH^•^ scavenging activity. This is consistent with the results in Figure 7a. When the polysaccharide concentration exceeded 3 mg/mL, the ability of P-CS to resist DPPH^•^ was improved significantly (Figure 7a), which might be due to the highest galacturonic acid (20.56%) (Table 5). The carboxyl group of galacturonic acid in polysaccharide may play the role of hydrogen-donating and electron-transfer to enhance the antioxidant effect [44]. Owing to the higher galacturonic acid content, the antioxidant activity of polysaccharide MFPs-90-40 from *Fructus Mori* was higher than MFP1P [45]. These results were consistent with our findings.

#### 3.6.2. ABTS^•+^ Scavenging Activity

ABTS^•+^ is one of the free radicals widely used in evaluating the antioxidant function of natural polysaccharides [46]. As seen in Figure 7b, the scavenging abilities of P-CL, P-CC and P-CS on ABTS^•+^ were dose-dependent. The IC_50_ values of P-CF, P-CL and P-CS were 0.45 ± 0.01, 0.94 ± 0.07, and 0.61 mg/mL ± 0.07, respectively. P-CF had the strongest scavenging activity of ABTS^•+^, which could reach 87.18% ± 0.72 at the concentration of 0.5 mg/mL. P-CC had the worst scavenging activity, and the ABTS^•+^ scavenging activity was only 21.58% ± 0.26 under the highest concentration (5 mg/mL). These results indicated that P-CF was beneficial to scavenging ABTS^•+^ to some extent.

The principle of ABTS^•+^ scavenging is that the hydroxyl and carboxyl groups in antioxidants provide electrons to convert free radicals into stable forms [46]. The significant scavenging effect of P-CF on ABTS^•+^ was related to molecular weight and monosaccharide composition. Hwang et al. obtained two polysaccharides with different molecular weights from persimmon and analyzed their antioxidant activity. The results showed that the polysaccharides with high molecular weight (>345 kDa) had stronger ability to resist ABTS^•+^ [47]. In addition, Wang et al. found that *Brassica rapa* L. polysaccharide with high molecular weight could scavenge more ABTS^•+^ [48]. Similar results were found in our study (Table 5), showing that P-CF had a higher molecular weight (128.06 kDa) with more significantly antioxidant activity. Furthermore, complex monosaccharide composition also has an effect on ABTS^•+^ scavenging. It is usually positively correlated with the content of galacturonic acid, galactose, arabinose, rhamnose, and mannose [39,40,43,44]. For instance, the polysaccharides that are rich in galactose and arabinose had stronger antioxidant activity to scavenge ABTS^•+^ [49]. Therefore, the antioxidant activity of polysaccharides are the results of multiple factors (molecular weight, monosaccharide composition, glycosidic bond type and linkage mode, chemical composition etc.), which needs further exploration.

#### 3.6.3. Hydroxyl Radical Scavenging Activity

Hydroxyl radical (•OH) is an extremely active reactive oxygen species, which can react with most biological macromolecules to detect whether they have the ability to resist free radicals [50]. The hydrogen provided by antioxidants is combined with the hydroxyl radical to terminate the free radical chain reaction and achieve the antioxidant effect. As shown in Figure 7c, with the increase in polysaccharide concentration, the free radical scavenging ability exhibited the fluctuating trends of *C. oleifera* polysaccharides. P-CF, P-CL, P-CC and P-CS reached the highest hydroxyl radical scavenging capacity at concentrations of 2, 4, 4 and 5 mg/mL, which were 79.97% ± 3.04, 65.24% ± 1.9, 63.1% ± 1.5, and 54.32% ± 2.68, respectively. P-CF had the best hydroxyl radical scavenging ability. This advantage may be related to molecular weight [47,48] and monosaccharide composition [39,43]. It was noteworthy that when the concentration increased from 1 to 4 mg/mL, the hydroxyl radical scavenging ability of P-CC was improved significantly. When the concentration exceeded 2 mg/mL, the free radical scavenging activity of P-CC was higher than that of P-CS. This may be attributed to P-CC having a triple helix structure (Figure 6), which makes it highly soluble in water and can improve antioxidant activity to some extent [51].

#### 3.6.4. Fe^2+^ Chelating Activity

Fe^2+^ is considered to be the most powerful and abundant lipid oxidation promoter in transition metals [52]. It can aggravate the damage to cells and become a thorny problem in food and other fields. Therefore, it is essential to take effective measures to chelate Fe^2+^. Our results showed that all of these polysaccharides exhibited dose-dependent (Fe^2+^) chelating activities (Figure 7d). However, this chelating effect was lower than that of EDTA-2Na. Maximum concentrations of P-CF, P-CL, P-CC, and P-CS that had the strongest Fe^2+^ chelating ability were 44.67% ± 1.04, 37.33% ± 0.53, 9.03% ± 0.82, and 10.27% ± 0.48, respectively. The abilities of P-CF and P-CL to chelate Fe^2+^ were stronger than those of P-CC and P-CS, which may be attributed to P-CF and P-CL having higher molecular weight (128.06 and 84.60 kDa) and galactose contents (23.78% and 29.59%). Sheng et al. showed that polysaccharide with higher molecular weight exhibited more significant reducing power [53]. Our results supposed that the higher molecular weight of polysaccharides may have more chelating groups, such as phenolic hydroxyl groups, which could chelate more Fe^2+^.

All of these four polysaccharides showed different antioxidant activities. This may be associated with the physicochemical properties and structural characteristics of different polysaccharides [54,55], such as monosaccharide composition, molecular weight, uronic acid content, glycosidic bond, group type, degree of polymerization, chain structure, and so on. Further experiments can start from the structure to further explore the structure–activity relationship of *C. Oleifera* polysaccharides.

## 4. Conclusions

In conclusion, the Box−Behnken design was used to optimize the extraction conditions of polysaccharides from *C. oleifera* by-products. The molecular weight and monosaccharide composition showed that the four polysaccharides were all heterogeneous polysaccharides, which were mostly composed of mannose, rhamnose, galacturonic acid, glucose, galactose, and xylose. Congo red tests showed that only P-CC had a triple helix structure. In addition, the antioxidant activity assays indicated that the four polysaccharides had potential in vitro free radical scavenging ability. Among them, P-CF had the strongest ability to scavenge DPPH^•^, ABTS^•+^, •OH. Therefore, these results suggested that the polysaccharides from *C. oleifera* could be applied to food antioxidants. The research on methylation analysis and nuclear magnetic resonance spectroscopy will be carried out in the future to further explain the structure of polysaccharides. Moreover, antioxidant activity will be further revealed in cell and in vivo experiments that can provide more favorable evidence for structure–activity relationship.

## Figures and Tables

**Figure 1 foods-11-03185-f001:**
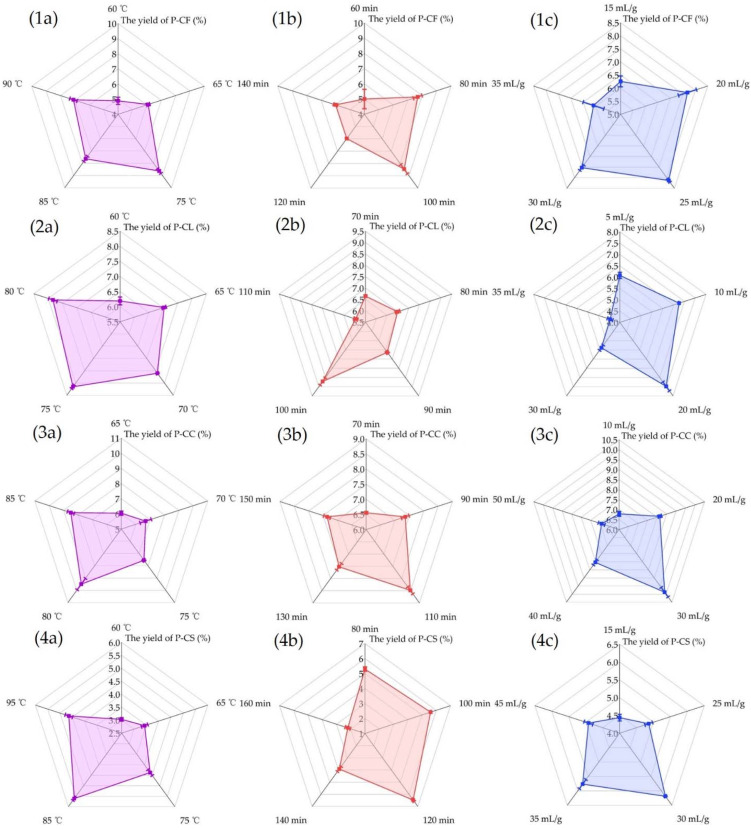
Effect of single factors on YP. Effect of (**a**) extraction temperature, (**b**) extraction time, and (**c**) liquid−solid ratio on YP (for (**1**) P-CF, (**2**) P-CL, (**3**) P-CC, (**4**) P-CS).

**Figure 2 foods-11-03185-f002:**
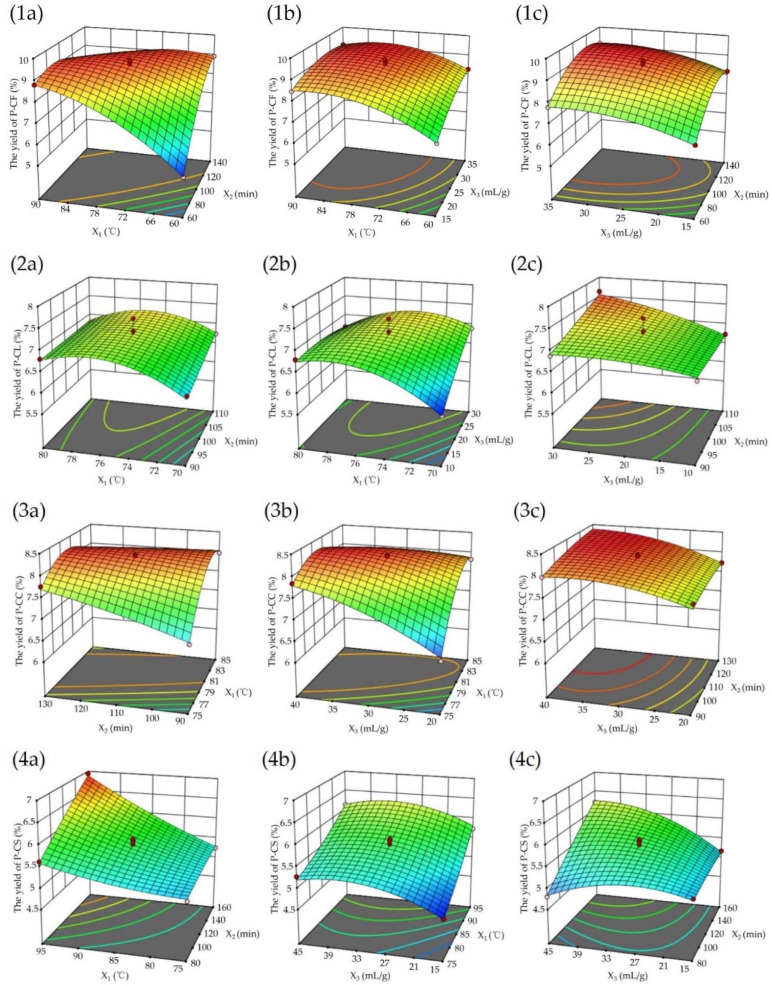
Results of response surface analysis. Response surface plots of the interactions between (**a**) extraction temperature and extraction time, (**b**) extraction temperature and liquid−solid ratio, (**c**) extraction time and liquid−solid ratio (for (**1**) P-CF, (**2**) P-CL, (**3**) P-CC, (**4**) P-CS). X_1_, X_2_, and X_3_ represent the extraction temperature, extraction time, and liquid−solid ratio, respectively.

**Figure 3 foods-11-03185-f003:**
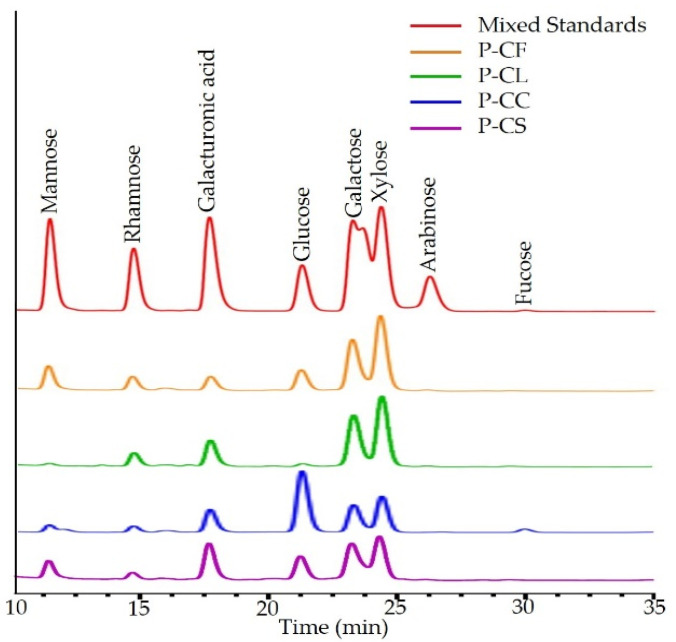
Monosaccharide composition of P-CF, P-CL, P-CC and P-CS.

**Figure 4 foods-11-03185-f004:**
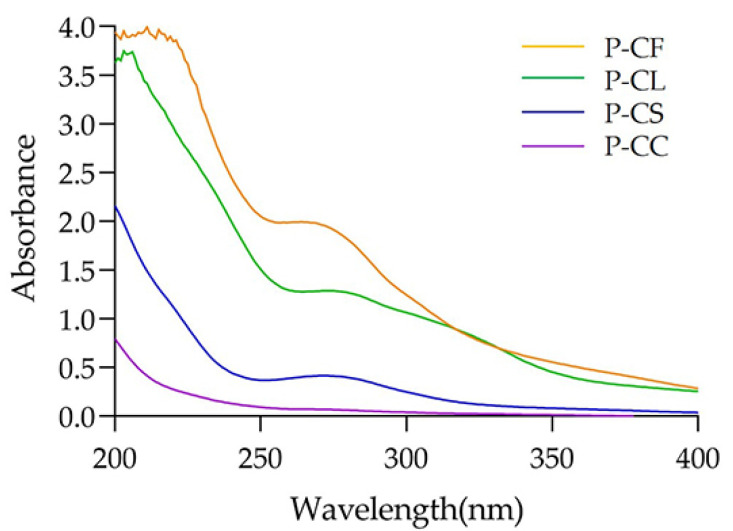
UV–vis spectra of P-CF, P-CL, P-CC and P-CS.

**Figure 5 foods-11-03185-f005:**
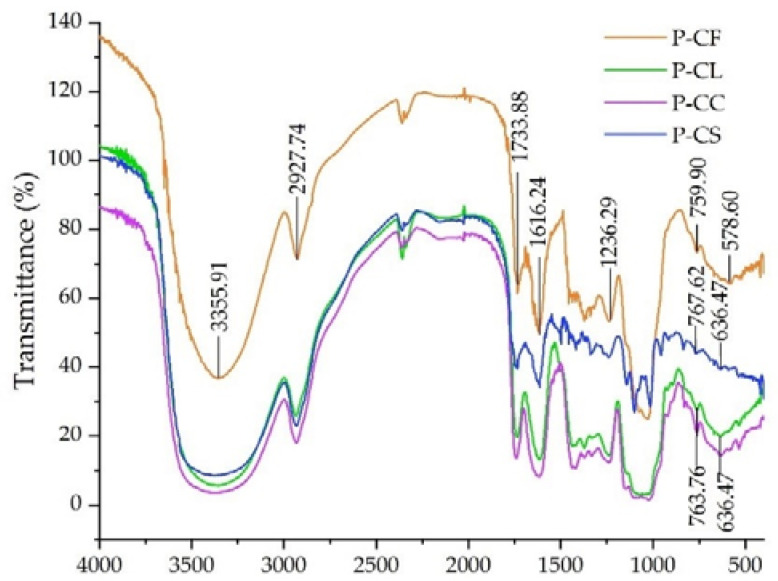
FT-IR spectra of P-CF, P-CL, P-CC, and P-CS.

**Figure 6 foods-11-03185-f006:**
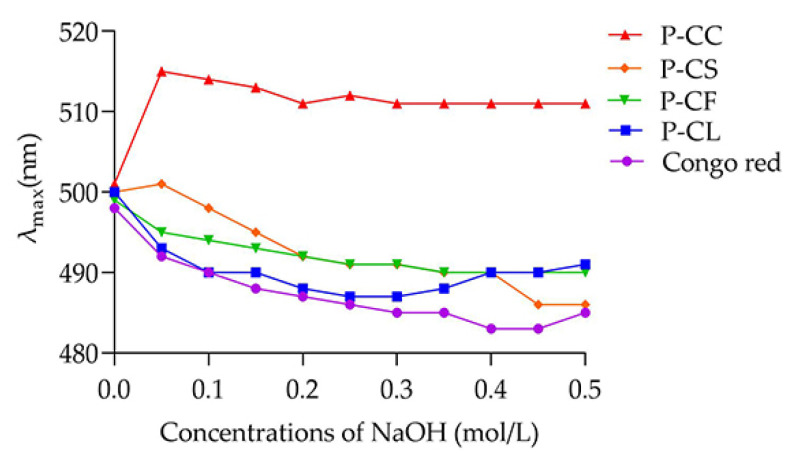
Cogon red test of P-CF, P-CL, P-CC, and P-CS.

**Figure 7 foods-11-03185-f007:**
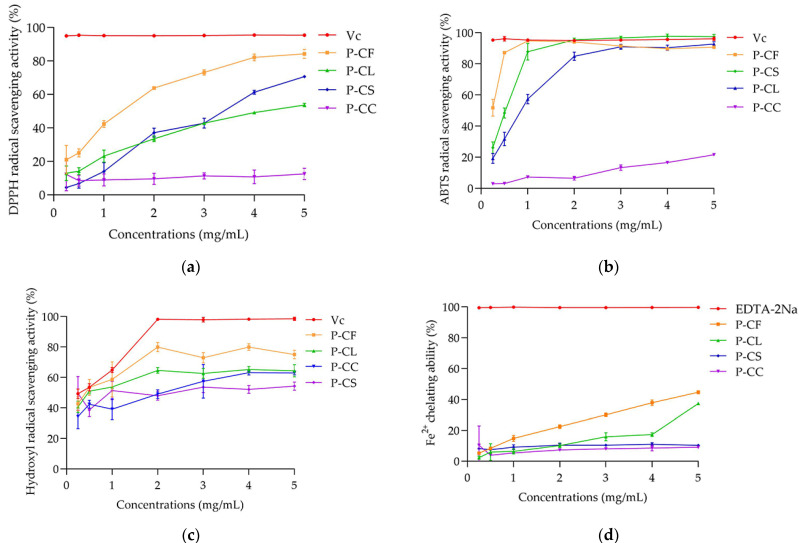
The antioxidant activity of P-CF, P-CL, P-CC, and P-CS. (**a**) DPPH^•^ scavenging activity; ABTS^•+^ scavenging activity (**b**); (**c**) hydroxyl radical scavenging activity; (**d**) Fe^2+^ chelating ability.

**Table 1 foods-11-03185-t001:** Determination of the level of independent variables parameters for BBD.

Level	Extraction Temperature (°C, X_1_)				Extraction Time (min, X_2_)				Liquid−Solid Ratio (mL/g, X_3_)			
	P-CF	P-CL	P-CC	P-CS	P-CF	P-CL	P-CC	P-CS	P-CF	P-CL	P-CC	P-CS
−1	60	70	75	75	60	90	90	80	15	10	20	15
0	75	75	80	85	100	100	110	120	25	20	30	30
1	90	80	85	95	140	110	130	160	35	30	40	45

**Table 2 foods-11-03185-t002:** Values of independent factors and response variables in BBD.

Run	Extraction Temperature (°C, X_1_)				Extraction Time (min, X_2_)				Liquid−Solid Ratio (mL/g, X_3_)				Polysaccharide Yield (%, Y)			
	P-CF	P-CL	P-CC	P-CS	P-CF	P-CL	P-CC	P-CS	P-CF	P-CL	P-CC	P-CS	P-CF	P-CL	P-CC	P-CS
1	0	0	1	0	0	0	0	0	0	0	1	0	9.35	6.84	7.71	5.82
2	1	0	0	0	0	−1	0	−1	1	1	0	−1	9.07	6.88	8.17	5.11
3	0	0	1	0	0	1	0	0	0	−1	−1	0	9.25	6.74	7.83	5.68
4	1	1	1	1	0	−1	−1	−1	−1	0	0	0	8.47	6.80	7.97	5.62
5	−1	0	0	0	0	1	−1	0	1	1	1	0	8.28	7.63	7.99	5.77
6	−1	−1	0	−1	−1	1	0	−1	0	0	0	0	5.24	6.75	8.19	5.05
7	1	0	−1	1	1	0	1	1	0	0	0	0	7.94	6.98	7.76	6.94
8	0	−1	0	0	−1	0	0	1	1	1	0	1	7.79	6.90	8.05	6.21
9	1	1	0	0	−1	0	0	−1	0	−1	0	1	8.81	6.80	8.21	4.80
10	0	1	0	1	1	0	1	0	−1	1	−1	1	8.20	6.69	7.74	6.13
11	0	0	−1	−1	0	−1	0	0	0	−1	1	−1	9.15	6.61	7.85	4.65
12	0	0	−1	0	1	0	0	1	1	0	−1	−1	9.15	7.44	6.40	5.17
13	−1	0	0	0	0	0	−1	0	−1	0	−1	0	6.78	6.90	7.67	5.73
14	0	−1	0	−1	0	0	1	1	0	−1	1	0	8.68	5.84	8.27	5.25
15	0	−1	0	1	0	−1	0	0	0	0	0	−1	8.98	6.28	8.11	5.73
16	−1	1	1	0	1	1	1	0	0	0	0	0	8.92	6.77	7.74	5.35
17	0	0	−1	−1	−1	0	−1	0	−1	0	0	1	6.77	7.13	6.76	5.28

**Table 3 foods-11-03185-t003:** Variance analysis results of regression models of the yields of P-CF, P-CL, P-CC, and P-CS.

		Mean Square		*F*-Value		*p*-Value		Significance	
Source	df	Y_1_	Y_2_	Y_1_	Y_2_	Y_1_	Y_2_	Y_1_	Y_2_
Model	9	2.17	0.25	54.48	6.60	<0.0001	0.0105	****	*
X_1_	1	3.21	0.21	80.76	5.57	<0.0001	0.0504	****	
X_2_	1	3.93	0.22	98.80	5.74	<0.0001	0.0478	****	*
X_3_	1	2.07	0.55	52.13	14.67	0.0002	0.0065	***	**
X_1_X_2_	1	5.20	0.06	130.77	1.64	<0.0001	0.2414	****	
X_1_X_3_	1	0.21	0.34	5.21	9.11	0.0564	0.0194		*
X_2_X_3_	1	<0.01	0.10	0.04	2.64	0.8562	0.1483		
X_1_^2^	1	1.47	0.70	36.91	18.46	0.0005	0.0036	***	**
X_2_^2^	1	2.46	<0.01	61.92	<0.01	0.0001	0.9867	***	
X_3_^2^	1	0.49	0.04	12.35	1.00	0.0098	0.3532	**	
Residual	7	0.04	0.04						
Lack of Fit	3	<0.01	0.01	0.02	0.20	0.9962	0.8917		
Pure Error	4	0.07	0.06						
Fit Statistics		Std. Dev.	Mean	C.V. %	R^2^	Adjusted R^2^	Predicted R^2^	Adequacy Precision	
	Y_1_	0.20	8.28	2.41	0.99	0.97	0.98	25.59	
	Y_2_	0.20	6.82	2.85	0.89	0.76	0.64	11.49	
**Source**	**df**	**Y_3_**	**Y_4_**	**Y_3_**	**Y_4_**	**Y_3_**	**Y_4_**	**Y_3_**	**Y_4_**
Model	9	0.43	0.54	44.62	23.77	<0.0001	0.0002	****	***
X_1_	1	0.76	2.21	78.59	96.30	<0.0001	<0.0001	****	****
X_2_	1	0.15	1.12	16.01	48.69	0.0052	0.0002	**	***
X_3_	1	0.59	0.39	61.12	16.87	0.0001	0.0045	***	**
X_1_X_2_	1	0.38	0.31	38.88	13.57	0.0004	0.0078	***	**
X_1_X_3_	1	0.61	0.01	62.85	0.59	<0.0001	0.4661	****	
X_2_X_3_	1	0.01	0.45	1.24	19.47	0.3030	0.0031		**
X_1_^2^	1	1.18	0.03	121.72	1.33	<0.0001	0.2871	****	
X_2_^2^	1	0.01	0.01	1.45	0.29	0.27	0.6099		
X_3_^2^	1	0.12	0.39	12.57	17.09	0.0094	0.0044	**	**
Residual	7	0.01	0.02						
Lack of Fit	3	0.02	0.01	3.98	0.23	0.1076	0.8683		
Pure Error	4	<0.01	0.03						
Fit Statistics		Std. Dev.	Mean	C.V. %	R^2^	Adjusted R^2^	Predicted R^2^	Adequacy Precision	
	Y_3_	0.10	7.79	1.26	0.98	0.96	0.79	25.28	
	Y_4_	0.15	5.55	2.73	0.97	0.93	0.88	19.35	

X_1_, X_2_, and X_3_ represent the extraction temperature, extraction time, and liquid−solid ratio, respectively. Y_1_, Y_2_, Y_3_, and Y_4_ represent the yields of P-CF, P-CL, P-CC, and P-CS, respectively. * denotes *p* < 0.05, ** denotes *p* < 0.01, *** denotes **** mean *p* < 0.001 and *p* < 0.0001, respectively.

**Table 4 foods-11-03185-t004:** Verifying the BBD model.

	Extraction Temperature (°C, X_1_)		Extraction Time (min, X_2_)		Liquid−Solid Ratio (mL/g, X_3_)		Polysaccharide Yield (%, Y)	
	Pre.	Pra.	Pre.	Pra.	Pre.	Pra.	Pre.	Pra.
P-CF	72.43	73	123.05	123	32.86	33	9.43	9.32 ± 0.11
P-CL	73.47	74	110	110	30	30	7.65	7.57 ± 0.11
P-CC	78.17	78	130	130	40	40	8.45	8.69 ± 0.16
P-CS	95	95	160	160	42.2	42	7.09	7.25 ± 0.07

Pre. and pra. represent the predicted value and the practical value.

**Table 5 foods-11-03185-t005:** Chemical component, molecular weight, and monosaccharide composition of P-CF, P-CL, P-CC, and P-CS.

Chemical Component	P-CF	P-CL	P-CC	P-CS
Total Carbohydrate (%)	59.48 ± 0.59	64.37 ± 0.66	50.72 ± 0.57	52.13 ± 0.16
Protein (%)	17.66 ± 0.1	11.77 ± 0.18	3.38 ± 0.2	7.74 ± 0.27
Mw Distribution (kDa)				
Peak 1	128.06	84.60	87.76	67.09
Peak 2	16.75	3.8	14.95	4.03
Peak 3				3.31
Component Ratio (%)	34.7:65.3	95.7:4.3	72.5:27.5	98.2:0.9:0.9
Monosaccharide Composition (mole%)				
Mannose	10.88	1	2.55	10.38
Rhamnose	6.34	7.07	3.35	3.78
Galacturonic Acid	5.92	12.8	11.86	20.56
Glucose	11.32	1.69	39.75	16.21
Galactose	23.78	29.59	15.82	20.76
Xylose	41.19	47.37	24.17	27.8
Arabinose	0.57	0.48		0.51
Fucose			2.5	

## Data Availability

The data set generated in this article can be provided to the corresponding author upon request.

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
