# Peer review of "Optimization of Extraction Process, Structure Characterization, and Antioxidant Activity of Polysaccharides from Different Parts of Camellia oleifera Abel"

_foods, 2022, doi:10.3390/foods11203185_

Round 1

Reviewer 1 Report

Comments in the document

Author Response

Response to Reviewer 1 Comments

Thanks for your review and comments. The following is my response to the comments.

Point 1: It is mentioned in the abstract that Check all DPPH and ABTS writing forms in the document and correct to DPPH and ABTS•+, respectively.

Response 1: We have checked all DPPH and ABTS writing forms, and corrected to DPPH and ABTS•+ in the manuscript. (Written in red in the manuscript.)

Point 2: In 2.1, Which season of the year was the material collected ?

Response 2: The C. oleifera leaves, flowers and fruits were collected in October. The by-products, including the seed cakes and fruit shells, were collected in November. All materials used in the article were collected in autumn. The ”in autumn” has been added in line 89.

Point 3: In 2.1, 40 mesh is the particle size of the material ?

Response 3: The 40 mesh is the sieve pore size during screening, rather than the particle size.

Point 4: In 3.1.1, Explain which structure and biological activity of polysaccharides are destroyed at high temperatures ?

Response 4: High-temperature treatment breaks the glycosidic bond of polysaccharides. The structure of polysaccharides is related to their biological activities. When the structure changes, its related biological activities (antioxidant, anti-inflammatory, hypoglycemic and so on) will also be affected. We have supplemented the content in the document in lines 199-202.

Point 5: In 3.1.1, Why when the liquid-solid ratio further increases, the yields of four polysaccharides began to decline ?

Response 5: This may be because the liquid-solid ratio reached a certain extent, and the polysaccharide has been extracted completely. Since then, continuously increasing the liquid-solid ratio will cause more impurities to dissolve into the extract, reducing the concentration of polysaccharides in the unit extract and bringing difficulties to the subsequent concentration work. We have supplemented the content in lines 226-230.

Point 6: In 3.1.4, Explain what changes in the structure and activity of polysaccharides under long time and high-temperature extraction ?

Response 6: Long time and high-temperature extraction treatment may break the glycosidic bond of polysaccharides. The structure of polysaccharides is related to their biological activities. When the structure changes, its related biological activities (antioxidant, anti-inflammatory, hypoglycemic and so on) will also be affected. We have supplemented the content in the document in lines 292-294.

Point 7: In 3.6, All this section can be improved including more references that support or explain more clear your result’s.

Response 7: All this section had been modified overall in lines 404-476.

Point 8: In conclusions, Explain why the structure-activity relationship of polysaccharides and their antioxidant activity in cells and in vivo can be discussed later?

Response 8: Since our study had confirmed that C. Oleifera polysaccharides had potential antioxidant activity in vitro. In order to further reveal the antioxidant activity and the possible structure-activity relationship of these polysaccharides, we will carry out cell and in vivo experiments. 

Reviewer 2 Report

The manuscript entitled "Optimization of Extraction Process, Structure Characterization and Antioxidant Activity of Polysaccharides from Different Parts of Camellia oleifera Abel" is a scientifically sound work done by the authors. There require extensive editing and modification prior to consideration of the manuscript. My specific comments are as follows;

1. In the abstract, apart from the percentage yield, I recommend to inlcude the varience (standard deviation/ error) while isolating the polysaccharide.

2. In the introduction part, authors detailed about the Camellia oleifera, however, description on the plant derived polysaccharides needs to be significantly improved. I recommend to include a description about plant polysaccharides and their biological and pharmacological significance using 3-4 references.

3. In methods section, I would like to know whether the species was taxonomically verified or any voucher specimen is deposited anywhere. This is a significant part.

4. How many times the authors extracted polysachharide? Is it a single time or multiple extractions were made. If so, provide the details of yield along with standard deviation.

5. GraphPad Prism software must include the country and state informaiton 

6. As the authors mentioned about DPPH radical scavenging, the other assays can also be mentioned briefly in the methodology.

7. Why the authors not estimated the IC50 value for DPPH and hydroxyl radical scavenging assays?

8. In conclusion section, I suggest to remove the data; instead, authors can include a description on future perspectives of the study.

Author Response

Response to Reviewer 2 Comments

Thanks for your review and comments. The following is my response to the comments.

Point 1: In the abstract, apart from the percentage yield, I recommend to inlcude the varience (standard deviation/ error) while isolating the polysaccharide.

Response 1: We have added the standard deviation in lines 19-20 and 26-27. (Written in red in the manuscript.)

Point 2: In the introduction part, authors detailed about the Camellia oleifera, however, description on the plant derived polysaccharides needs to be significantly improved. I recommend to include a description about plant polysaccharides and their biological and pharmacological significance using 3-4 references.

Response 2: We have added five references in lines 67-74 to introduce the biological and pharmacological significance of plant polysaccharides.

Point 3: In methods section, I would like to know whether the species was taxonomically verified or any voucher specimen is deposited anywhere. This is a significant part.

Response 3: All experimental materials in this study were from Camellia oleifera. Moreover, all materials were identified by Professor Chunbang Ding (College of Life Sciences, Sichuan Agricultural University). We have supplemented the content in lines 91-92.

Point 4:  How many times the authors extracted polysachharide? Is it a single time or multiple extractions were made. If so, provide the details of yield along with standard deviation.

Response 4: Based on Box-Behnken Design, the polysaccharides in this study were all extracted at single time, and the yields were 8.15% (P-CF), 7.02% (P-CL), 7.77% (P-CC) and 6.72% (P-CS) , respectively.

Point 5: GraphPad Prism software must include the country and state informaiton.

Response 5: We have added country and state information for GraphPad Prism software in line 184.

Point 6:  As the authors mentioned about DPPH radical scavenging, the other assays can also be mentioned briefly in the methodology.

Response 6: We have supplemented the determination methods of antioxidant activity in lines 170-179.

Point 7: Why the authors not estimated the IC50 value for DPPH and hydroxyl radical scavenging assays?

Response 7: Since the ability of P-CC and P-CS to scavenging DPPH and four polysaccharides to scavenging hydroxyl radicals exhibited a fluctuating trend, it is difficult to estimate their accurate IC50 values. For P-CF and P-CL, their IC50 values for scavenging DPPH were: 1.53 ± 0.26 and 3.16 ± 0.75. We have supplemented the content in lines 399-400.

Point 8: In conclusion section, I suggest to remove the data; instead, authors can include a description on future perspectives of the study.

Response 8: We have removed the data and added a description of the prospect in lines 489-501. 

Reviewer 3 Report

- L33- change "less in" to "also found in".

- Introduction contains minimal details regarding the extraction and characterization process, please elaborate.

- Author should provide details of how these polysaccharides express antioxidant activities in the introduction part.

- L75 - is there any specific collection procedure? or its crude samples?

- L79 - I recommend "chemicals and reagents"

- Although, BBD would choose a best condition, author still have the input base range for temperatures, extraction time, and L-S ratio, right? please include those in the section 2.3.

- L108 - Please elaborate the methods.

- What is the purpose of measuring 2.5.4?

- L137 - please elaborate the methods.

- Section 2.5.6, 2.6, please elaborate the methods.

-2.7 - Statistical analysis, lack of analysis details, please include.

- L322 - I think this section can be removed from the manuscript.

- L169 -Not sufficient explanation, please add more.

- L183- degradation of polysaccharides to lower molecule weight? please link your monosaccharide results to this discussion.

- Section 3.5 - It’s not clear, is author trying to express the results based on the extractions, or with NaOH, a complete rewrite required for this section.

-L368- explain more why PCF has more antioxidant activity as compared with the other samples, provide direct explanation., Similar comments applicable to other antioxidant activities as well.

Author Response

Response to Reviewer 3 Comments

Thanks for your review and comments. The following is my response to the comments.

Point 1: L33- change "less in" to "also found in".

Response 1: We have corrected "less in" to "also found in" in lines 34-35. (Written in red in the manuscript.)

Point 2: Introduction contains minimal details regarding the extraction and characterization process, please elaborate.

Response 2: We have added a detailed description of the extraction and characterization process to the introduction in lines 77-82.

Point 3: Author should provide details of how these polysaccharides express antioxidant activities in the introduction part.

Response 3: We have supplemented the details of how these polysaccharides express biological activity in the introduction in lines 48-51, 55-58 and 59-61.

Point 4: L75 - is there any specific collection procedure? or its crude samples?

Response 4: The flowers and leaves were collected from C. oleifera. The shells and seed cakes were collected before and after oil extraction.

Point 5: L79 - I recommend "chemicals and reagents"

Response 5: We have corrected " Chemicals " to " Chemicals and Reagents " in line 94.

Point 6: Although, BBD would choose a best condition, author still have the input base range for temperatures, extraction time, and L-S ratio, right? please include those in the section 2.3.

Response 6: We also took that into account. However, due to the large number of samples involved in this study, if this part is added, the manuscript will look redundant. In order to make up this deficiency, we presented the range of extraction temperature, time and liquid-solid ratio in Table 1 and Figure 1.

Point 7: L108 - Please elaborate the methods

Response 7: We have supplemented the method of chemical composition determination in lines 124-126 and 128-130.

Point 8: What is the purpose of measuring 2.5.4?

Response 8: UV-visible spectroscopy can detect the purity of polysaccharides. If there are impurities such as protein and nucleic acid in polysaccharides, there will be peaks at 260 and 280 nm. Since our polysaccharides contain a small amount of protein, UV-visible spectroscopy can perform the most basic characterization of polysaccharides, such as the characteristic peaks of polysaccharides and the characteristic peaks of protein and nucleic acid.

Point 9: L137 - please elaborate the methods.

Response 9: We have supplemented the fourier transform infrared (FTIR) spectroscopy method in lines 156-157.

Point 10: Section 2.5.6, 2.6, please elaborate the methods.

Response 10: We have supplemented the determination methods of Congo red and antioxidant activity in lines 160-166 and 170-179.

Point 11: 2.7 Statistical analysis, lack of analysis details, please include.

Response 11: We have supplemented the details of statistical analysis in lines 181-186.

Point 12: L322 - I think this section can be removed from the manuscript.

Response 12: UV-visible spectroscopy can detect the purity of polysaccharides. If there are impurities such as protein and nucleic acid in polysaccharides, there will be peaks at 260 and 280 nm. since our polysaccharides contain a small amount of protein, UV-Visible Spectroscopy can perform the most basic characterization of polysaccharides, such as the characteristic peaks of polysaccharides and the characteristic peaks of nucleic acids and proteins. We think this part can be presented.

Point 13: L169 -Not sufficient explanation, please add more.

Response 13: We have added explanations for the decrease in yield due to higher extraction temperature in lines 199-202.

Point 14: L183- degradation of polysaccharides to lower molecule weight? please link your monosaccharide results to this discussion.

Response 14: Our statement in this section is to explain the reason for the decrease in polysaccharide yield caused by longer extraction time.

Point 15: Section 3.5 - It’s not clear, is author trying to express the results based on the extractions, or with NaOH, a complete rewrite required for this section.

Response 15: We 've rewritten the Congo red analysis in lines 381-388.

Point 16: L368- explain more why PCF has more antioxidant activity as compared with the other samples, provide direct explanation., Similar comments applicable to other antioxidant activities as well.

Response 16: We have supplemented the analysis of DPPH results in lines 404-408 and 413-420.

Round 2

Reviewer 2 Report

The manuscript has been improved significantly and therefore I accept the manuscript in the present form.

Reviewer 3 Report

The present version of the paper can be accepted.